# TRAF3 gene regulates macrophage migration and activation by lung epithelial cells infected with *Aspergillus fumigatus*

Shumi Shang,[1] Dan He,[1] Cong Liu,[2] Xinyuan Bao,[1] Shuaishuai Han,[3] Li Wang[1]

**ABSTRACT** *Aspergillus fumigatus* is an opportunistic pathogen that infects immuno-compromised patients and imposes a heavy burden on global health. Searching for key genes in host resistance to *A. fumigatus* infection can help us understand the mechanisms of *A. fumigatus*-host interactions and provide new targets for the treatment of fungal infections. TRAF3 is one of the intracellular adapter proteins of the innate immune response that regulates signaling in various cellular processes, including host defense against pathogens. However, the defense mechanism of TRAF3 against *A. fumigatus* infection remains unknown. In this study, we found that TRAF3 overexpression led to the adhesion and internalization of more spores of *A. fumigatus* in lung epithelial cells and thus greater host immune surveillance evasion. Meanwhile, TRAF3 was able to regulate the expression of pro-inflammatory cytokines in lung epithelial cells infected with *A. fumigatus* through the negative regulation of NF-κB and MAPK signaling pathways. In this study, we also examined the effects of TRAF3 in the interaction between lung epithelial cells, macrophages, and *A. fumigatus* spores, and the results showed that TRAF3-overexpressing lung epithelial cells reduced the migration and activation of macrophages after *A. fumigatus* infection. *In vivo* experiments using TRAF3-overexpressing transgenic zebrafish larvae revealed that TRAF3 overexpression increased the fungal load and mortality of zebrafish infected with *A. fumigatus*. In conclusion, the TRAF3 gene can negatively regulate the resistance of lung epithelial cells to *A. fumigatus*, which plays an important role in the early infection processes of *A. fumigatus*.

**IMPORTANCE** *Aspergillus fumigatus* can infect immunocompromised individuals and cause chronic and fatal invasive fungal infections. A better understanding of the molecular mechanisms of *A. fumigatus*-host interactions may provide new references for disease treatment. In this study, we demonstrated that the TRAF3 gene plays an important role in the early infection of *A. fumigatus* by regulating the resistance of lung epithelial cells to *A. fumigatus*. Macrophages are the most abundant innate immune cells in the alveoli; however, few studies have reported on the interactions between lung epithelial cells and macrophages in response to *A. fumigatus* invasion. In our study, it was demonstrated that the TRAF3 gene reduces migration to macrophages and cytokine production by negatively regulating lung epithelial cell adhesion and internalization of *A. fumigatus* spores. Together, our results provide new insights into lung epithelial cell-macrophage interactions during *A. fumigatus* infection.

**KEYWORDS** TRAF3, *Aspergillus fumigatus*, lung epithelial cells, macrophage, zebrafish

*A*spergillus fumigatus infection of immunocompromised patients (e.g., those with hereditary immunodeficiency, those who use immunosuppressive drugs, and transplant and AIDS patients) causes invasive aspergillosis (IA). According to reports, more than 300,000 new cases of invasive aspergillosis are reported each year worldwide, with a mortality rate of 30% to 95% (1). On 25 October 2022, *A. fumigatus* was listed in

Address correspondence to Li Wang, wli99@jlu.edu.cn.

The authors declare no conflict of interest.

See the funding table on p. 15.

the critical priority group of the World Health Organization fungal priority pathogens list (2). *A. fumigatus* is widely present in the natural environment, and humans inhale large numbers of *A. fumigatus* spores every day. However, *A. fumigatus* infections rarely occur in immunocompetent individuals, suggesting that the body's immune system is capable of recognizing and clearing *A. fumigatus* spores to avoid infection (3, 4).

Lung epithelial cells are situated within the alveoli and cover 95% of the alveolar surface and are organisms' first line of defense against *A. fumigatus* spores (5). In immunocompetent hosts infected with *A. fumigatus*, lung epithelial cells provide a physical barrier and participate in the clearance of spores using mucosal cilia, preventing spores from colonizing the lungs. However, in immunocompromised patients, lung epithelial cells have a reduced ability to clear spores, leading to increased adhesion and germination of *A. fumigatus* (6). Some studies have shown that *A. fumigatus* infects lung epithelial cells *in vitro*, first adhering to the cell surface, followed by internalization into the cell, and after 6 h of infection, its spores begin to germinate budding tubes, known as small hyphae, which in turn cause an invasive infection and recruit professional phagocytes such as macrophages and neutrophils to the site of infection (7, 8).

Alveolar macrophages make up 5% of the total cells in the alveoli, making them the primary resident innate immune cells in the alveoli. Their major role is to phagocytize and kill *A. fumigatus* spores through oxidative mechanisms and non-oxidative mechanisms of phagosomal acidification and to release cytokines and chemokines to regulate the inflammatory response (9–11). Lung epithelial cells and alveolar macrophages coexist in the alveoli and are capable of participating together in the immune response when exposed to inhalation stimuli (12). However, there are few reports on the influence of lung epithelial cells on macrophages in the host response to *A. fumigatus* infection.

Tumor necrosis factor receptor-associated factor 3 (TRAF3), a member of the TRAF family with E3 ubiquitin ligase activity, is a key node of innate and adaptive immune receptor signaling. It has been shown that TRAF3 is a versatile adapter protein that can participate in the formation of multiple protein complexes and can positively regulate the production of type I interferon while negatively regulating MAPK and NF-κB signaling pathways (13, 14). It has been demonstrated that TRAF3 plays a role in the development of various inflammatory and pathogenic microbial infections, but the function of TRAF3 in anti-infective immunity to *A. fumigatus* is not clear (15, 16).

A detailed understanding of the molecular mechanisms of *A. fumigatus*-host interactions could help provide new insights into how to treat *A. fumigatus*. In this study, we investigated the function of TRAF3 in the early stages of infection with *A. fumigatus* using lung epithelial cells (A549) *in vitro*. Furthermore, macrophages were added to the *A. fumigatus*-lung epithelial cell co-culture model to analyze the effects of changes in TRAF3 gene expression in lung epithelial cells on macrophage migration and cytokine expression. This study provides a new reference for the molecular mechanisms behind the interactions between *A. fumigatus*, lung epithelial cells, and macrophages.

## RESULTS

### Expression of TRAF3 in lung epithelial cells

To investigate the role of the TRAF3 gene in *A. fumigatus* infection of lung epithelial cells, the expression levels of TRAF3 were analyzed after 6 h of infection. The results of qRT-PCR and western blot analysis showed a significant decrease in TRAF3 mRNA and protein expression levels in A549 cells infected with *A. fumigatus* compared to the control group (Fig. 1A through C). Immunofluorescence analysis was conducted to detect the expression and localization of TRAF3 in A549 cells. The results showed that TRAF3 was expressed in both the nucleus and cytoplasm. However, when the cells were stimulated by *A. fumigatus*, the expression of TRAF3 was significantly decreased (Fig. 1D and E). These results suggest that the TRAF3 gene is sensitive to early infection by *A. fumigatus*, and lung epithelial cells respond to early infection by *A. fumigatus* by down-regulating TRAF3 expression.

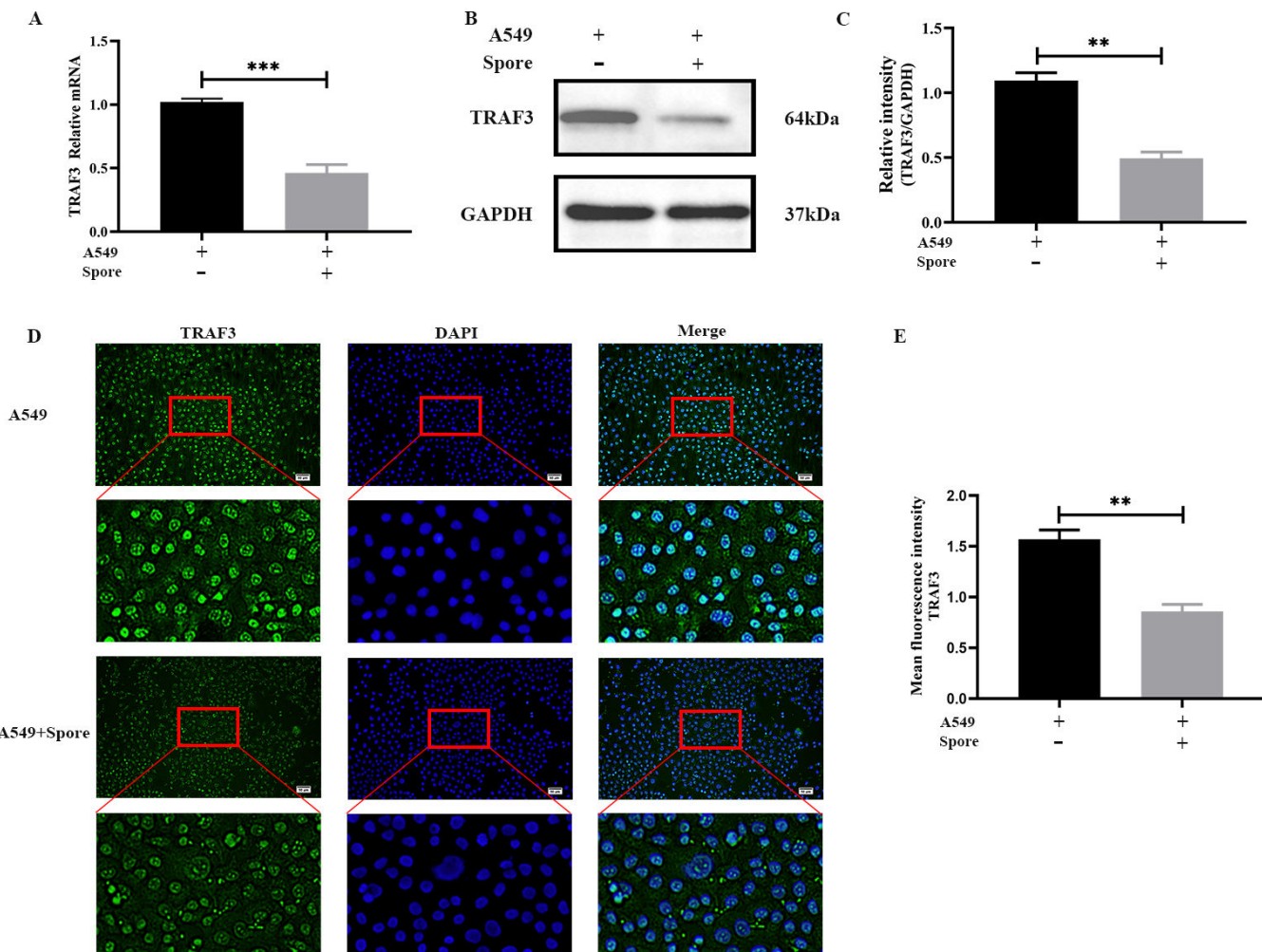

**FIG 1** Lung epithelial cell TRAF3 expression is down-regulated under *A. fumigatus* infection. Expression of TRAF3 was detected in A549 cells stimulated or unstimulated with *A. fumigatus* by qRT-PCR and western blot (A, B, C). The expression and localization of TRAF3 in A549 cells were determined by immunofluorescence (D, E). Results are expressed as the mean ± SD from three independent experiments. **$P < 0.01$, ***$P < 0.001$. Scale bars: 50 µm.

## Effect of TRAF3 on adhesion and internalization of *A. fumigatus* spores in lung epithelial cells

To study the role of TRAF3 in early infection with *A. fumigatus*, a pIRES-TRAF3-EGFP plasmid was used to induce TRAF3 overexpression in A549 cells, and pIRES-EGFP plasmid transfection was used as a control. Compared with the control cells, A549 cells transfected with the TRAF3 overexpression plasmid showed an increased expression of TRAF3 by at least 140-fold at the mRNA level (Fig. 2A) and nearly 2.5-fold at the protein level (Fig. 2B and C).

The number of cells adhering to *A. fumigatus* spores was measured, and the results showed that TRAF3 overexpression significantly increased the ability of A549 to adhere to *A. fumigatus* spores (Fig. 2D and E). Subsequently, this study examined the effect of TRAF3 on the internalization of *A. fumigatus* spores by A549 cells, and the results of the nystatin protection assay showed that TRAF3 overexpression increased the internalization rate of A549 cells to 30% compared with a rate of 24% in the control group (Fig. 2F).

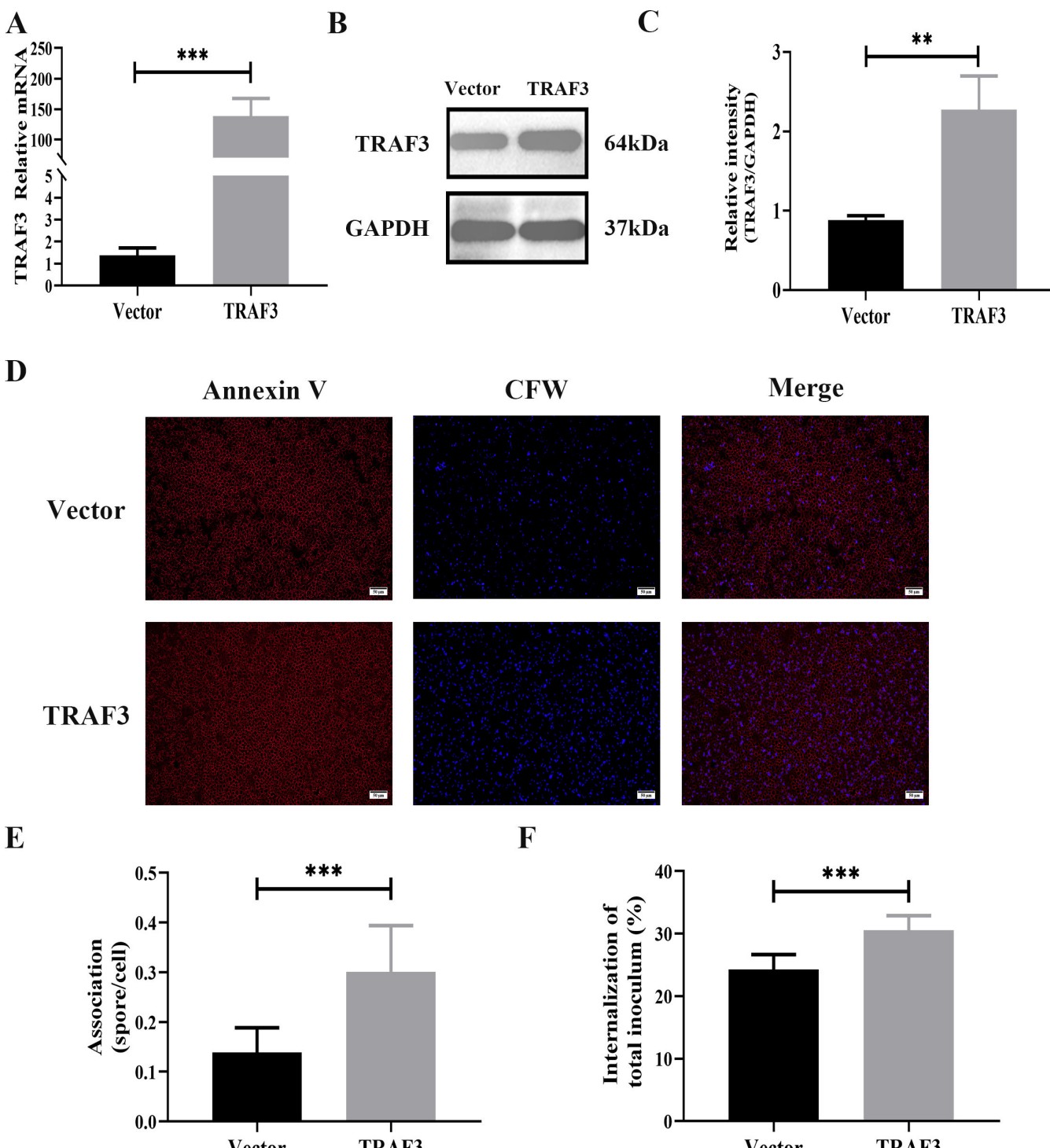

**FIG 2** TRAF3 overexpression promotes lung epithelial cell adhesion and internalization of *A. fumigatus* spores. A549 cells were transfected with pIRES-TRAF3-EGFP plasmid (TRAF3) and control pIRES-EGFP plasmid (Vector). mRNA (A) and protein (B, C) expression of TRAF3 were determined by qRT-PCR and western blot. *A. fumigatus* spores were co-incubated with A549 cells with or without TRAF3 overexpression for 6 h. *A. fumigatus* spores were labeled with fluorescent white (CFW), then fixed with 4% paraformaldehyde. Cells were dead after fixation and then Annexin-V-stained cell membrane; the number of *A. fumigatus* spores adhered to lung epithelial cells was determined by fluorescence (D). (E) is the number of *A. fumigatus* spores adhered to each cell. The number of internalized *A. fumigatus* spores with or without TRAF3 overexpression in A549 cells was determined by nystatin protection assay (F). Results are expressed as the mean ± SD of three independent experiments. **$P < 0.01$, ***$P < 0.001$. Scale bars: 50 µm.

## Effect of TRAF3 on the inflammatory response induced by *A. fumigatus* infection of lung epithelial cells

We further evaluated the function of TRAF3 in controlling the inflammatory response of lung epithelial cells to *A. fumigatus* infection. We stably overexpressed TRAF3 in lung epithelial cells, A549, with or without *A. fumigatus* stimulation and detected changes in IL-1β, TNF-α, IL-6, and IL-8 mRNA levels in the A549 cells by quantitative real-time PCR and the secretion of the cytokines by ELISA. The results showed that the mRNA and protein levels of these pro-inflammatory cytokines were significantly up-regulated by *A. fumigatus* stimulation, regardless of whether the TRAF3 gene was overexpressed in A549 cells. Meanwhile, TRAF3 overexpression in A549 cells did not enhance the basal production of these cytokines but was able to inhibit the increase in mRNA and protein levels of pro-inflammatory cytokines induced by *A. fumigatus* (Fig. 3). These results indicate that TRAF3 inhibits the *A. fumigatus*-induced production of pro-inflammatory cytokines.

## Role of TRAF3 in the interaction between lung epithelial cells and macrophages in *A. fumigatus* infection

During the early stages of *A. fumigatus* infection, macrophages are key phagocytes required for the clearance of *A. fumigatus* spores (17). To assess whether TRAF3 plays a role in lung epithelial cell interactions with macrophages after infection with *A. fumigatus*, we constructed an *in vitro A. fumigatus* spore-A549-macrophage co-culture model (Fig. 4A). The results showed that TRAF3 overexpression in A549 cells inhibited the migration of macrophages upon *A. fumigatus* infection (Fig. 4B and C) and inhibited cytokine production by macrophages (Fig. 4D). In contrast, conditioned medium collected from *A. fumigatus* co-cultured with TRAF3 overexpression or control A549 cells failed to promote the migration of macrophages and had no obvious effect on their cytokine expression (Fig. S1). These results suggest that the migration of macrophages and the induction of cytokine expression require the presence of *A. fumigatus* spores. Furthermore, to confirm that macrophage migration was not due to direct injury to lung epithelial cells, cell viability was assessed by detecting 3-(4,5-dimethyl-2-thiazolyl)-2,5-diphenyl-2H-tetrazolium bromide conversion. And cell membrane integrity was assessed by measuring lactate dehydrogenase release (18). The results showed that cell viability

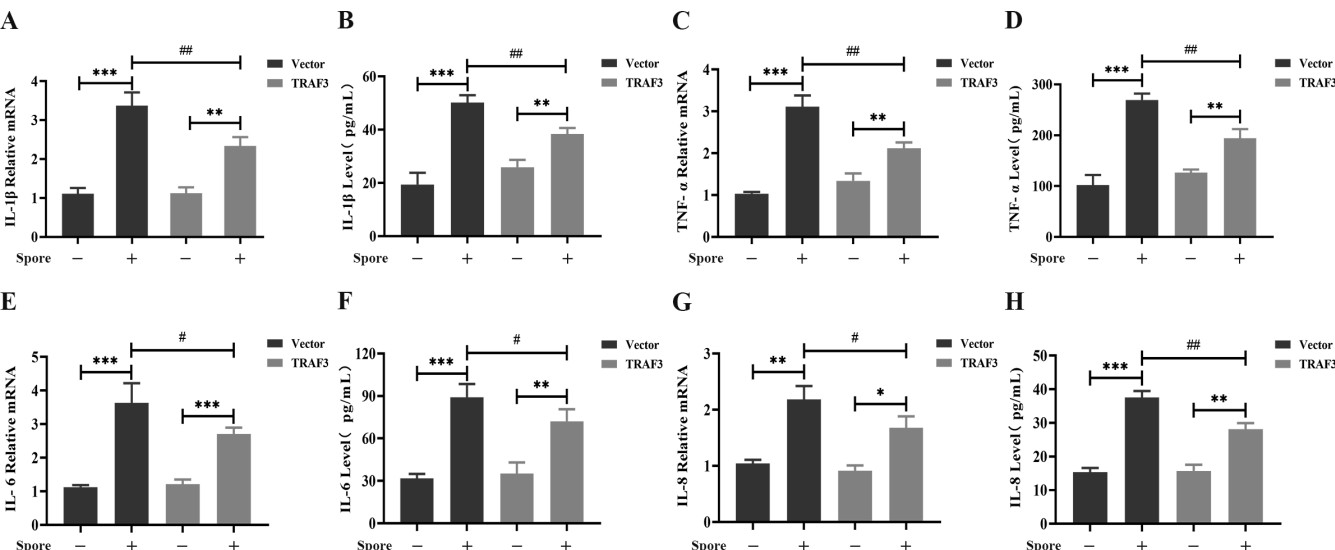

**FIG 3** TRAF3 inhibits the inflammatory response of *A. fumigatus*-stimulated activated lung epithelial cells. IL-1β (A), TNF-α (C), IL-6 (E), IL-8 (G) mRNA levels were detected by quantitative real-time PCR in A549 cells stably overexpressing TRAF3 or vector control with or without *A. fumigatus* stimulation for 6 h; the secretion of IL-1β (B), TNF-α (D), IL-6 (F), and IL-8 (H) was detected by ELISA in A549 cell culture supernatants. Results are expressed as the mean ± SD of three independent experiments. *, # $P < 0.05$; **, ## $P < 0.01$; ***$P < 0.001$.

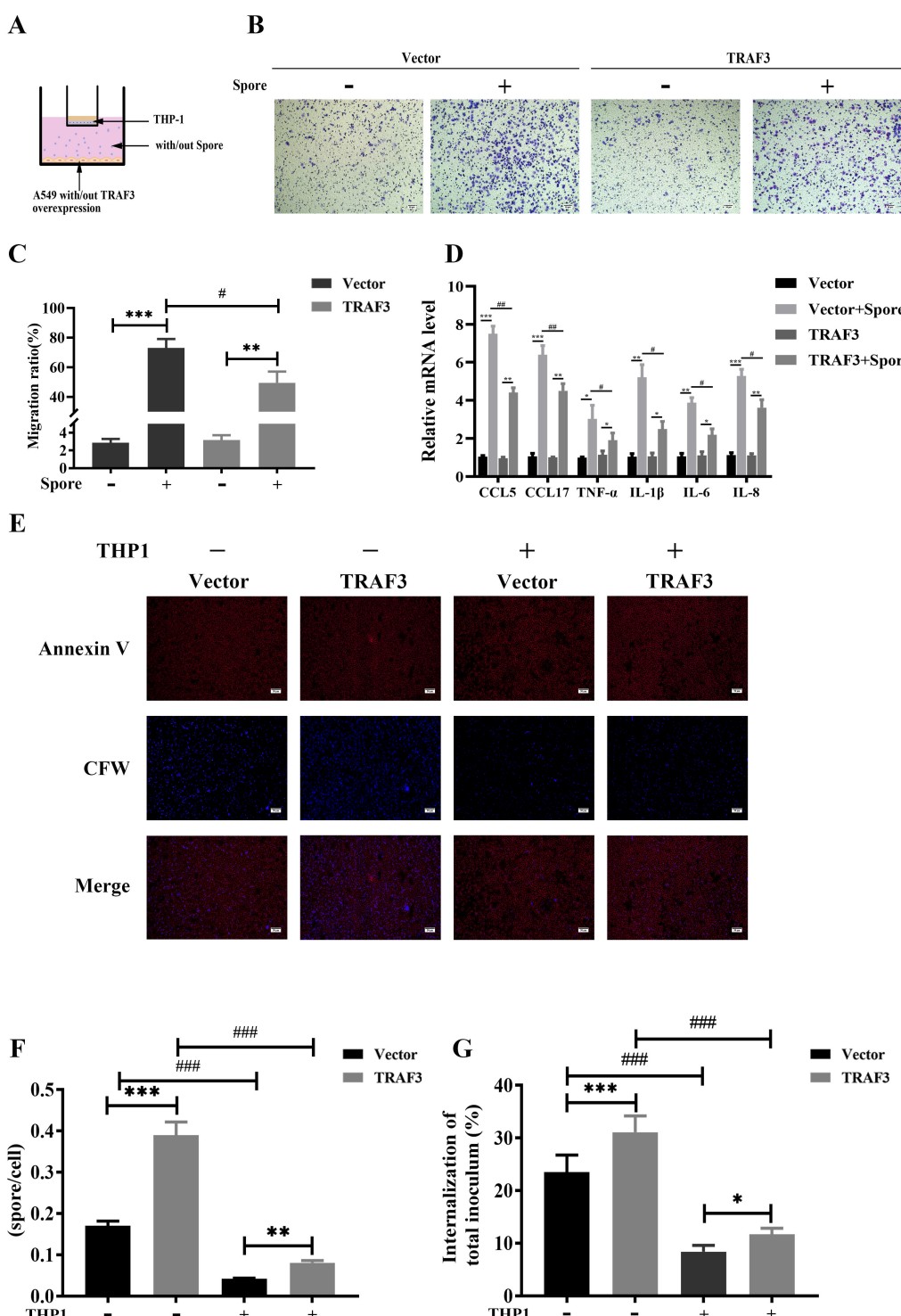

**FIG 4** TRAF3 inhibits macrophage migration and macrophage cytokine production by lung epithelial cells infected with *A. fumigatus*. (A) Schematic of co-culture of *A. fumigatus*, lung epithelial cells, and macrophages. Macrophages that migrated to the lower chamber were fixed with 4% paraformaldehyde, stained with Giemsa staining solution, and counted (B, C). The expression of cytokines in the macrophages in the upper chamber of Transwell was analyzed by quantitative real-time PCR (D). Fluorescent staining and nystatin protection assay were performed on lung epithelial cells at the bottom of 24-well plates to detect A549 cells adhesion (E, F) and internalization of *A. fumigatus* spores (G). Results are expressed as the mean ± SD of three independent experiments. *, # $P < 0.05$; **, ## $P < 0.01$; ***, ### $P < 0.001$. Scale bars: 50 µm.

was significantly decreased, but cell membrane integrity was not altered after 6 h of *A. fumigatus* infection of A549 cells, regardless of whether TRAF3 was overexpressed. Moreover, TRAF3 overexpression in A549 cells did not affect cell viability and cell membrane integrity (Fig. S2).

The effect of macrophages on lung epithelial cell adhesion and the internalization of spores were examined by fluorescent staining and nystatin protection assay. The results showed that the presence of macrophages reduced the adhesion ratio of A549 cells with *A. fumigatus* spores from 0.17 to 0.04 and the internalization rate from 23.5% to 8.4%, while the adhesion ratio of TRAF3-overexpressing A549 cells with *A. fumigatus* spores was reduced from 0.39 to 0.08 and the internalization rate reduced from 31.1% to 11.7% (Fig. 4E through G). These results indicate that macrophages were able to significantly inhibit the adhesion and internalization of most spores to A549 cells, and TRAF3 overexpression was able to enhance the adhesion and internalization of A549 cells to *A. fumigatus,* regardless of the presence of macrophages.

## Effect of TRAF3 on NF-κB and MAPK signaling pathways in *A. fumigatus*-induced lung epithelial cells

To explore the molecular mechanisms by which TRAF3 regulates lung epithelial cell adhesion, internalizes *A. fumigatus* spores, and produces inflammatory factors, we analyzed the effects of TRAF3 on MAPK and NF-κB signaling pathways in lung epithelial cells. The results showed that A549 cells stimulation by *A. fumigatus* significantly promoted the phosphorylation of p65 and IκBα, but the phosphorylation of p65 and IκBα was inhibited by TRAF3 overexpression in A549 cells infected with *A. fumigatus* (Fig. 5A and B). The p65 nuclear translocation is key to activation of the NF-κB signaling pathway. Overexpression in TRAF3 reduced the accumulation of p65 in the nuclei of *A. fumigatus*-stimulated induced A549 cells (Fig. 5C and D). Furthermore, immunofluorescence analysis showed that TRAF3 overexpression in A549 cells inhibited the *A. fumigatus*-induced nuclear translocation of p65 (Fig. 5E). Detection of MAPK signaling pathway-related proteins showed that TRAF3 overexpression decreased the expression of JNK, p-ERK, and p-AKT induced by *A. fumigatus* in A549 cells (Fig. 5F and G).

## Role of TRAF3 in zebrafish resistance to *A. fumigatus* infection

In order to investigate the role played by TRAF3 during *A. fumigatus* infection, transgenic zebrafish overexpressing TRAF3 were established, and zebrafish larvae were selected to be infected with *A. fumigatus* by immersion method (19). The Tol2 transposable plasmid with ubiquitous expression of green fluorescent protein was mixed with transposase mRNA at a mass ratio of 1:1 and microinjected into zebrafish one-cell stage embryos. Transgenic zebrafish with TRAF3 overexpression were screened by fluorescence microscopy (Fig. 6A). The results of qRT-PCR showed that the expression of TRAF3 was significantly decreased according to the time of infection of zebrafish larvae with *A. fumigatus* (Fig. 6B). Survival analysis showed that the survival rate of TRAF3-overexpressing transgenic zebrafish (31%) after infection was significantly lower than that of wild-type zebrafish (51%) (Fig. 6C). In addition, the fungal load results showed that the number of spores swallowed by TRAF3-overexpressing transgenic zebrafish larvae immersed in E2 medium containing *A. fumigatus* spores was not significantly different from that of wild-type zebrafish within 2 h, indicating that the infection model was successfully established. After transferring the infected zebrafish to sterile E2 medium, the fungal load in TRAF3-overexpressing transgenic zebrafish was significantly higher than that in wild-type zebrafish at all time points post-infection as infection time increased (Fig. 6D).

Pro-inflammatory cytokines are rapidly produced in response to infection and promote autoimmune inflammation in the organism. Therefore, qRT-PCR was used to examine the expression of relevant pro-inflammatory cytokines in zebrafish larvae at various time points after infection with *A. fumigatus*. The results showed that the expression of IL-1β, TNF-α, IL-6, and IL-8 pro-inflammatory cytokines was significantly

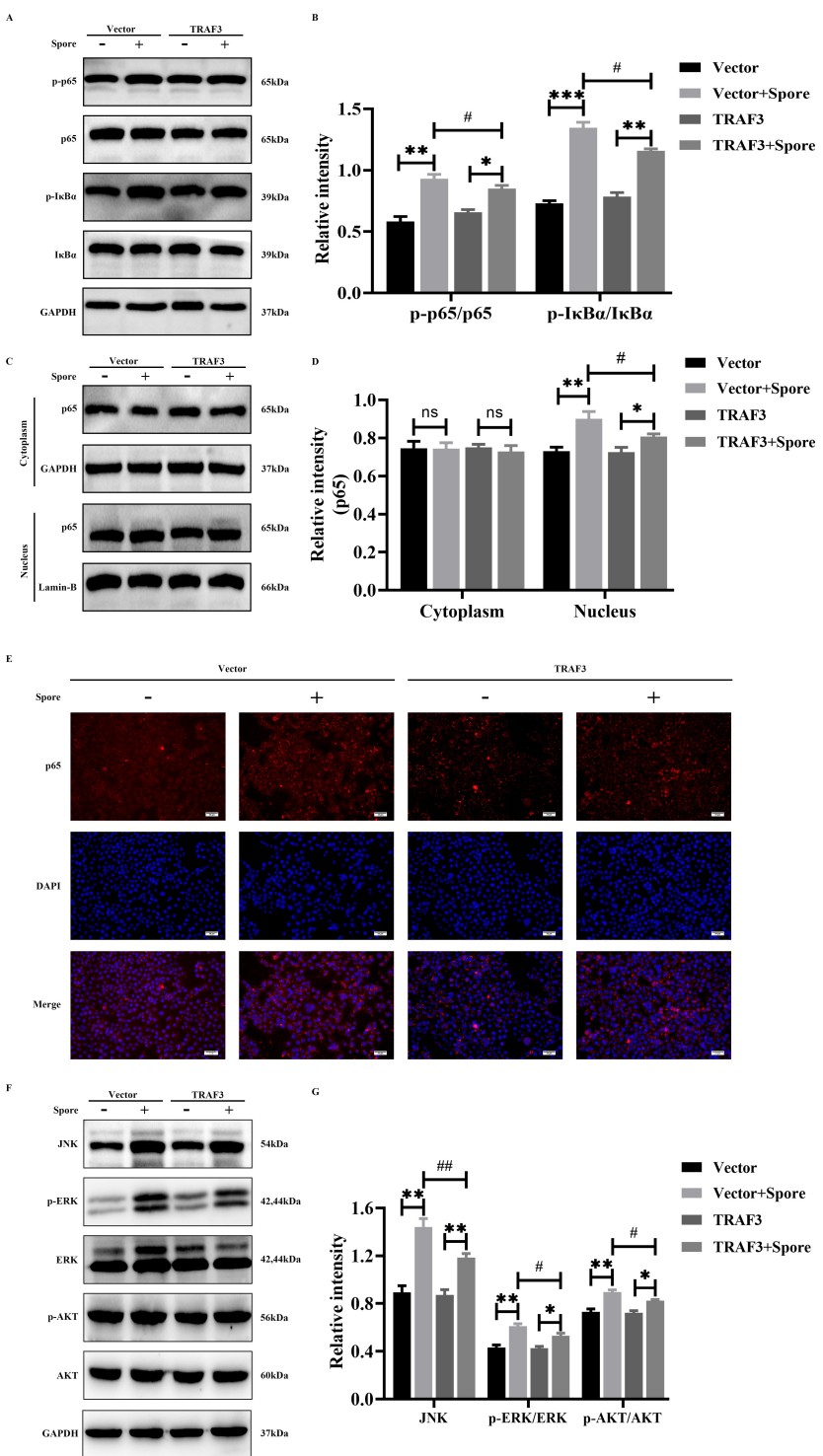

**FIG 5** TRAF3 overexpression inhibits the activation of NF-κB and MAPK signaling pathways in *A. fumigatus*-infected lung epithelial cells. Protein expression of p-p65, p65, p-IκBα, IκBα, JNK, p-ERK, ERK, p-AKT, and AKT was detected by western blot for 6 h in A549 cells stimulated with or without TRAF3 overexpression by *A. fumigatus* (A, B, F, G). The protein expression of p65 in the nucleus and cytoplasm of A549 cells was detected by western blotting (C, D). The expression and localization of p65 in A549 cells were determined by immunofluorescence (E). Results are expressed as the mean ± SD of three independent experiments. ns stands for no statistically significant difference; *, # $P < 0.05$; **, ## $P < 0.01$; ***$P < 0.001$. Scale bars: 50 μm.

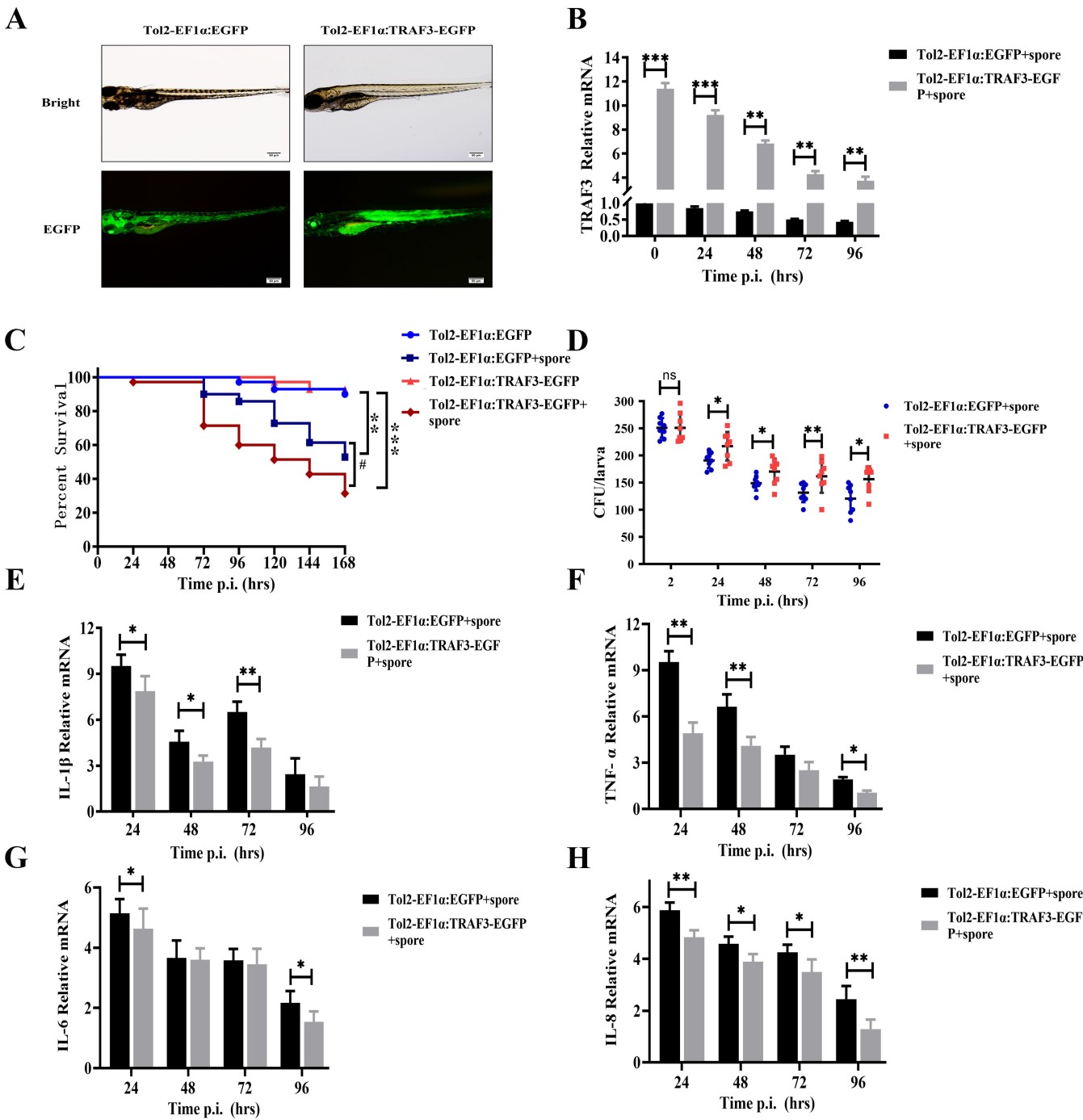

**FIG 6** TRAF3 overexpression reduces survival and promotes fungal load *in vivo* in *A. fumigatus*-infected zebrafish. (A) Images of zebrafish larvae transgenic for TRAF3 and expressing EGFP at 3 dpf. qRT-PCR was used to detect the transcriptional expression levels of TRAF3 at various time points after zebrafish infection with *A. fumigatus* (B). Zebrafish larvae were immersed in E2 medium containing $1 \times 10^7$ *A. fumigatus* spores and replaced with sterile E2 medium after 2 h. The effect of TRAF3 on the survival rate of zebrafish larvae infected with *A. fumigatus* (C) and the fungal load of zebrafish larvae at various time points (D) were examined. The expression levels of pro-inflammatory cytokines (E–H) in zebrafish infected with *A. fumigatus* at various time points were measured by qRT-PCR. ns stands for no statistically significant difference; *, # $P < 0.05$; ** $P < 0.01$; *** $P < 0.001$. Scale bars: 50 µm.

down-regulated in TRAF3-overexpressing transgenic zebrafish compared with wild-type zebrafish (Fig. 6E through H).

## DISCUSSION

*A. fumigatus* infection can lead to an inflammatory response in the host (20). Research has indicated that TRAF3 plays an active role in inflammation-related diseases such as multiple myeloma, systemic lupus erythematosus, and herpes simplex encephalitis (16, 21, 22). TRAF3 is a key node of innate and adaptive immune receptor signaling and is primarily involved in regulating inflammation, antiviral responses, and cell apoptosis (13). However, the impact of TRAF3 on host resistance to *A. fumigatus* infection and its related mechanisms remain unclear and worth exploring.

The clearance of *A. fumigatus* spores from the lungs is linked to the activation of the inflammatory response and secretion of pro-inflammatory cytokines (23). Research has shown that TRAF3 knockout mice die in the early stages of life, indicating that this gene has an important biological function in postnatal development and in maintaining normal immune system function. Currently, there are no agonists or inhibitors available to investigate the TRAF3 gene (24, 25). This study demonstrated that TRAF3 expression decreased after *A. fumigatus* infection of lung epithelial cells. Therefore, experiments were designed for the overexpression of the TRAF3 gene *in vivo* and *in vitro* models to, in turn, demonstrate the function played by the TRAF3 gene during the early infection of *A. fumigatus*. Furthermore, TRAF3-overexpressing lung epithelial cells showed increased adherence to and internalization of *A. fumigatus* spores, but their expression of pro-inflammatory cytokines was significantly suppressed. This suggests that TRAF3 has an active role in the host immune response to *A. fumigatus* infection.

Lung epithelial cells and other stromal cells can adhere to and internalize *A. fumigatus* spores, but not all the spores are killed, and they can remain latent within the cells, where they evade immune recognition, thus increasing the chance of *A. fumigatus* spore germination and infection (26, 27). In this study, we found that TRAF3-overexpressing A549 cells adhered to and internalized more *A. fumigatus* spores than control A549 cells; however, they induced a 23.5% decrease in the migration rate of macrophages and a decrease in the expression of macrophage cytokines. Therefore, it was hypothesized that the internalization of *A. fumigatus* spores by lung epithelial cells increased, leading to reduced exposure of the spores to immune cells. Macrophage recognition of *A. fumigatus* spores was therefore reduced, reducing macrophage migration (Fig. 7). Additionally, the study found that medium from a co-culture of A549 cells and *A. fumigatus* failed to induce macrophage migration, suggesting that the TRAF3-induced pro-inflammatory cytokines expressed by lung epithelial cells infected with *A. fumigatus* exerted less influence on lung epithelial cell-macrophage interactions.

Through bioinformatics analysis of TRAF3 gene-related signaling pathways, it was discovered that TRAF3 is located upstream of the NF-κB and MAPK signaling pathways. The activation of NF-κB and MAPK signaling pathways is essential for the production of *A. fumigatus*-induced inflammation in epithelial cells. The activation of these two signaling pathways is mainly associated with the expression of Dectin-1 and Sky genes (7, 28, 29). In this study, it was found that TRAF3 could inhibit the activation of NF-κB and MAPK signaling pathways, thus reducing the release of epithelial cell inflammatory factors induced by *A. fumigatus*. Therefore, it is speculated that the TRAF3 gene plays a key role in initiating the activation of related signaling pathways after *A. fumigatus* infection of epithelial cells.

Animal models used to study pathogenic microbial infestations and infections include nematodes, mice, rabbits, zebrafish, etc. Compared to nematodes, zebrafish possess more of the basic characteristics of the human immune system (30). Compared to animal models such as mice and rabbits, zebrafish are genetically easier to manipulate, optically transparent, and possess multiple functional organ systems early in development (31). Therefore, the zebrafish is an ideal model for studying *A. fumigatus*-host interactions. In this study, we took advantage of the properties of zebrafish to construct a transgenic zebrafish with TRAF3 systemic overexpression and determined the success of the transgenic zebrafish construction by detecting fluorescent protein expression, which saved a lot of time and money in the study.

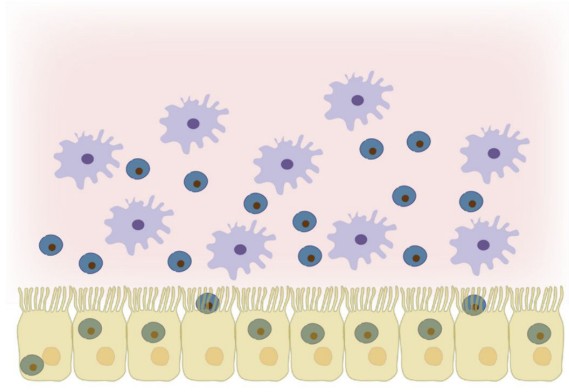
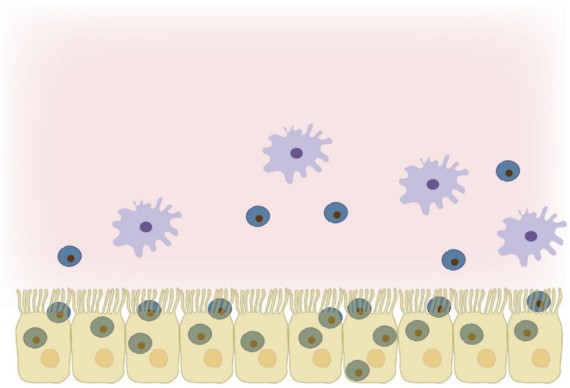

**lung epithelial cells**

**TRAF3 overexpression in lung epithelial cells**

## TRAF3 overexpressing lung epithelial cells can harbor more *Aspergillus fumigatus* spores and consequently reduce the migration of macrophages

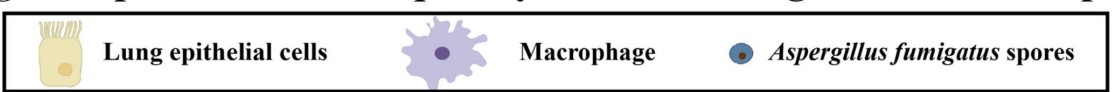

**Lung epithelial cells** **Macrophage** *Aspergillus fumigatus* **spores**

**FIG 7** Schematic representation of epithelial cells infected with *A. fumigatus* affecting the migration of macrophages. (Left) When lung epithelial cells are infected with *A. fumigatus* spores, a large number of macrophages migrate to the infection site. (Right) TRAF3 overexpression causes lung epithelial cells to adhere to and internalize a large number of *A. fumigatus* spores; fewer *A. fumigatus* spores are recognized by macrophages, and only a small number of macrophages migrate to the infection site.

The construction of *A. fumigatus*-infected zebrafish models was performed by microinjection of *A. fumigatus* spores into the hindbrain ventricle (32). *A. fumigatus* often infects the lungs through the respiratory tract in clinical practice, thus by referring to the construction of a bacterial infection model by Laanto et al., we used a static immersion method to establish a zebrafish model of *A. fumigatus* infection in this study (23, 33). Zebrafish can be infected with *A. fumigatus* spores when they open their mouths and gills during respiratory movements, and their infection mode is more likely to mimic clinical infection. In addition, this method is simple to operate and requires less instrumentation than the microinjection method of infection. The fungal load experiment in this study showed that the infection model established by the static immersion method was stable and reproducible. It is suggested that this infection method can not only be used to construct fungal infection models but also for drug development and testing.

In conclusion, this study highlights the significance of the TRAF3 gene in lung epithelial cells during early *A. fumigatus* infections. This gene negatively regulates NF-κB and MAPK signaling pathways, which in turn affect the expression of pro-inflammatory cytokines in lung epithelial cells after *A. fumigatus* infection. Additionally, TRAF3 influences macrophage migration and related cytokine expression by modifying epithelial cell adhesion and internalizing *A. fumigatus* spores. These findings suggest that the TRAF3 gene is a potential therapeutic target for diseases associated with *A. fumigatus* infection.

## MATERIALS AND METHODS

### *A. fumigatus* culture

The *A. fumigatus* strain IFM40808 used in this study was a clinical isolate gifted from the Medical Mycology Research Center of Chiba University, Japan, and was stored in the laboratory. *A. fumigatus* strains were grown on potato dextrose agar (PDA) slants and at

37°C for 4 days. The slants were rinsed with saline containing 0.01% Tween 20, the spore suspension was collected, and the hyphae were filtered out using a sterile cell strainer and counted using a hemocytometer.

## Cell cultures

Human lung type II lung epithelial cells (A549) and human monocytes (THP-1) used for these experiments were purchased from the American Type Culture Collection. The cells were grown in RPMI-1640 medium (Gibco; Thermo Fisher Scientific, Inc.) containing 10% fetal bovine serum (Gibco; Thermo Fisher Scientific, Inc.) and cultured in a 37°C, 5% $CO_2$ cell culture incubator. THP-1 cells were derived from patients with acute monocytic leukemia and could differentiate into macrophage-like cells by exposure to phorbol 12-myristate 13-acetate (PMA). Before the experiment, THP-1 cells were induced with 100 ng/mL PMA (Absin, Shanghai, China) for 72 h to differentiate into THP-1-like macrophages (34).

## Cell transfection

The TRAF3 overexpression plasmid pIRES-EGFP (pIRES-EGFP-CMV-MCS-EGFP-Neo) used for cell transfection was purchased from Beijing Syngentech Co., Ltd., China. A549 cells were seeded into well plates and cultured overnight, washed three times with RPMI-1640 medium, and transfected according to the instructions for Lipofectamine 3000 transfection reagent (Thermo Fisher Scientific). The enhanced fluorescence of green fluorescent protein was observed with an Olympus Model IX71 fluorescent microscope (Evident Olympus, Tokyo, Japan) at 48 h post-transfection.

## Internalization

Internalization of *A. fumigatus* spores by A549 was assayed using the nystatin protection assay (35). A549 cells were seeded at $5 \times 10^4$ cells/well into 24-well plates (Corning, Costar, New York, USA) and cultivated overnight in a cell incubator at 37°C with 5% $CO_2$. *A. fumigatus* spores were co-cultured with A549 cells at MOI = 10:1 for 6 h. After discarding the supernatant, the cells were washed three times with PBS, and RPMI-1640 medium containing 20 µg/mL nystatin was added and incubated for 4 h to kill extracellular spores. Cells were then washed three times with PBS and incubated with 0.25% Triton X-100 (Boster Biotechnology, China) for 15 min to lyse the cells. The lysates were diluted 100-fold and inoculated onto PDA plates, and individual colony-forming units (CFUs) were counted after culture for 48 h in a 37°C incubator. The spore internalization rate was calculated with this formula: Internalization rate = intracellular conidia colonies/initial inoculum of conidia × 100% (36).

## Adherence

Adhesion values represent the percentage of total conidia bound to A549 cells (26, 37). Coverslips (15 mm) (Corning, Costar, New York, USA) were put in the bottom of 24-well plates, which were subsequently seeded with A549 cells. After the cells were adhered with 90% confluence, *A. fumigatus* spores were cultured with the A549 cells at MOI = 10:1 for 6 h in a cell incubator at 37°C with 5% $CO_2$. The cells were washed three times with PBS to remove spores that were not bound to the cells. Adherent spores were labeled with 0.003% Calcofluor-white (Sigma, USA) and incubated for 5 min in the dark. Then, the cells were washed three times with PBS, and 4% paraformaldehyde was added for 20 min at room temperature to fix the cells. After the cells were fixed, the cell membranes were labeled with Annexin V-PE staining solution (Beyotime Biotechnology, China) and incubated for 10 min in the dark. Excess staining solution was washed off with PBS, and the coverslips removed from the wells. Subsequently, coverslips were mounted on glass slides using Anti-Fluorescence Attenuating Sealer (Solarbio Science & Technology, China). In order to determine the total number of spores adhered to lung epithelial cells, 12 randomly selected fields of the coverslips were photographed with a BX53

microscope (Evident Olympus, Tokyo, Japan). The number of spores adhered to A549 cells was counted using Image J software.

## ELISA

TRAF3-overexpression or control A549 cells were co-cultured with *A. fumigatus* spores for 6 h. The supernatant was removed and centrifuged at 4℃, 5,000 rpm for 5 min. The supernatant was transferred into a new 1.5-mL EP tube. Levels of IL-1β, TNF-α, IL-6, and IL-8 in supernatants were measured by ELISA kits (Servicebio, Wuhan, China) according to the reagent manufacturer's protocol.

Cells were washed three times with pre-cooled PBS, lysed in 6-well plates with radioactive immunoprecipitation assay buffer containing 1 mM phenylmethylsulfonyl fluoride, and centrifuged at 12,000 rpm for 20 min, and the supernatant was aspirated and used to detect total cellular protein. Nuclear and Cytoplasmic Protein Extraction Kit was purchased from Absin, Shanghai, China. Protein concentration was measured via BCA Protein Assay Kit (Beyotime Biotechnology, China). The 10% SDS-PAGE gels were prepared using the PAGE Gel Rapid Preparation Kit (Shanghai Epizyme Biomedical Technology Co., Ltd). Then, 15 µg of sample protein was loaded into each well for electrophoresis. Subsequently, the proteins were transferred onto polyvinylidene difluoride (PVDF) membranes (Millipore, USA), blocked with Western high-performance blocking solution (Genefirst, Shanghai, China) for 10 min, washed three times with TBST for 5 min each time, and incubated with primary antibody overnight at 4℃; then the wash steps were repeated, and the cells were incubated with secondary antibody at room temperature (Absin, Shanghai China) for 2 h. After repeating the washing steps, immunoblots were developed with enhanced chemiluminescence ECL kit (Biosharp, China) detection reagents, and images were acquired using a Tanon 4200 chemiluminescence imaging system (Tanon, Shanghai, China). The gray values of the images were analyzed with ImageJ software. The antibodies p65 (1:1,000), IκBα (1:2,000), JNK (1:1,000), p-ERK (1:2,000), ERK (1:2,000), AKT (1:1,000), and GAPDH (1:2,000) used in this experiment were purchased from PTM Biolabs, Hangzhou, China. The antibodies TRAF3 (1:1,000), p-p65 (1:1,000), p-IκBα (1:1,000), p-AKT (1:1,000), and LaminB (1:1,000) used in this experiment were purchased from Wanleibio, Shenyang, China.

## Western blotting

### Polymerase chain reaction

For cellular RNA extraction, 400 µL TRIzol was added to the cell culture wells for lysis. For zebrafish RNA extraction, 10 zebrafish were placed into 1.5-mL EP tubes, and 400 µL TRIzol added for lysis (38). RNA was isolated and purified according to the manufacturer's instructions for the use of TRIzol reagent (TaKaRa, Japan). RNA was reverse transcribed into cDNA using a cDNA Reverse Transcription Kit (Monad, Shanghai, China). Real-time quantitative PCR (RT-qPCR) analysis was performed using SYBR Green PCR Mix (Monad, Shanghai, China) in the ABI QuantStudio 3 PCR system (Applied Biosystems, Waltham, MA, USA). The primers used are listed in Table 1.

### Macrophage migration and co-culture in Transwell

TRAF3-overexpressing A549 or control A549 cells with *A. fumigatus* spores were seeded into the lower chambers of Transwell plates (Corning, Costar, New York, USA) containing polyethylene terephthalate membrane inserts with a pore size of 5 µm (or 0.4 µm). The THP-1-like macrophages were digested, centrifuged, and resuspended in RPMI-1640, and the cell concentration was adjusted to $1 \times 10^6$ cells/mL. Then, 200 µL of cells was inoculated into the upper chambers of Transwell plates, which were cultured for a total of 6 h at 37℃ in a 5% $CO_2$ cell culture incubator. Cells in the upper chamber of 5 µm Transwell plates were fixed with 4% paraformaldehyde and stained with Giemsa staining solution (Beyotime Biotechnology, China). The suspended cells on the upper

**TABLE 1** Primers used for qPCR

| Human | Forward | Reverse |
|---|---|---|
| TRAF3 | TGACCAGATGCTGAGTGTGC | CTGGCTGTAAAGGGACAGGG |
| IL-1β | ACGATGCACCTGTACGATCA | TCTTTCAACACGCAGGACAG |
| TNF-α | AACCTCCTCTCTGCCATCAA | CCAAAGTAGACCTGCCCAGA |
| IL-6 | TACCCCCAGGAGAAGATTCC | TTTTCTGCCAGTGCCTCTTT |
| IL-8 | TGTGAAGGTGCAGTTTTGCC | ACCCAGTTTTCCTTGGGGTC |
| GAPDH | CAAATTCCATGGCACCGTCA | GGACTCCACGACGTACTCAG |
| CCL5 | ATGACTCCCGGCTGAACAAG | GCCTCCCAAGCTAGGACAAG |
| CCL17 | ACTTCAAGGGAGCCATTCCC | TGTTGGGGTCCGAACAGATG |
| **Zebrafish** | **Forward** | **Reverse** |
| TRAF3 | GCAGGTCATGGAGCATTTGG | AGGACAGACGGTTTCTTTGTGT |
| IL-1β | TTTGTGGGAGACAGACGGTG | TCAGGGCGATGATGACGTTC |
| TNF-α | AGACCTTAGACTGGAGAGATGAC | CAAAGACACCTGGCTGTAGAC |
| IL-6 | GCAGTATGGGGGGAACTATCCG | TCCTGACCCCTTCAAATGCC |
| IL-8 | TTGAAACAGAAAGCCGACGC | CTTAACCCATGGAGCAGAGGG |
| EF1α | AAGCTTGAAGACAACCCCAAGAGC | ACTCCTTTAATCACTCCCACCGCA |

layer of the polyethylene terephthalate membrane were cleaned out, photographed with an IX71 microscope, and analyzed with Image J software to calculate the number of migrated cells. Cells in the upper chamber of 0.4-µm Transwell plates were harvested for quantitative polymerase chain reaction, and A549 cells in the lower chamber were used for the detection of conidia in internalization and adhesion assays.

## Immunofluorescence

A549 cells were seeded into 24-well plates containing 15-mm coverslips and co-cultured with *A. fumigatus* spores for 6 h. Cells were fixed with 4% paraformaldehyde and washed three times with PBS for 5 min each. We added 0.2% Triton-100 (Boster Biotechnology, China) to break down the membranes at room temperature for 30 min. The washing step was repeated, and 10% albumin (Solarbio Science & Technology, China) added and incubated for 1 h at room temperature. The washing step was repeated, and the primary antibody added and incubated overnight at 4°C. The washing step was repeated, and the secondary antibody added and incubated at room temperature for 1 h. Then 30 µL of anti-fluorescence quenching blocker containing DAPI was added (Absin, Shanghai, China) dropwise to the coverslip; the coverslip was removed from the well plate, the surface with the cells side placed onto a coverslip, and pictures were taken with a fluorescence microscope BX53. Primary antibodies, such as TRAF3 (1:50) was purchased from Wanleibio, Shenyang, China, and p65(1:50) was purchased from PTM Biolabs, Hangzhou, China. Secondary Cy3-labeled goat anti-mouse IgG antibodies were purchased from Servicebio, Wuhan, China; FITC-labeled goat anti-mouse IgG antibodies were purchased from Absin, Shanghai, China.

## Transgenic zebrafish construction

Wild-type AB zebrafish (ZFIN ID: ZDB-GENO-960809-7) were used in this study, and all experiments were approved by the Ethics Committee of Jilin University. The Tol2-EF1α:EGFP and Tol2-EF1α:TRAF3-EGFP plasmids and Tol2 transposase messenger RNA used in this study were purchased from Hunter Biotech, China. Transposase mRNA was mixed with Tol2-EF1α:EGFP or Tol2-EF1α:TRAF3-EGFP plasmids in equal proportions (39). Zebrafish embryos were arranged on a mold and injected into animal poles with a glass needle under a microscope (ASI MPPI-3). Injected zebrafish embryos were cultured in a constant temperature incubator at 28°C with 14-h/10-h light-dark cycle. Injected zebrafish embryos were cultured in a constant temperature incubator at 28°C with 14-h/ 10-h light and darkness. Zebrafish embryos 3 days post fertilization were screened for EGFP fluorescence, indicating the expression of transgenic lines.

## Static immersion infection model

The *A. fumigatus* spore suspension was prepared at a concentration of $10^7$ CFU/mL. Zebrafish larvae at 3 days post fertilization were placed in a 96-well plate with one fish per well. Spore suspension (100 µL) was added to the well plate containing the zebrafish and soaked for 2 h. The spore suspension was replaced with E2 culture medium, and the survival rate was recorded every 24 h.

## Fungal burden determination

Six zebrafish were homogenized in sterile E2 culture medium. The homogenate was diluted 10-fold with PBS, and 50 µL was spread onto PDA plates and incubated in an incubator at 37°C for 48 h. The fungal burden of zebrafish was calculated by counting CFU.

## Statistical analysis

At least three independent replicates of each experiment were performed in this study, and data are expressed as mean ± standard deviation. Statistical analyses were performed using GraphPad Prism software version 8.0 (Dotmatics, San Diego, CA, USA). An unpaired two-tailed *t*-tests were used for comparisons between two groups. And one-way ANOVA followed by *t*-test was used for comparison between three or more groups. $P < 0.05$ was considered to indicate a statistically significant difference.

### ACKNOWLEDGMENTS

We thank Suzanne Leech, Ph.D., from Edanz for editing a draft of this manuscript.

This study was supported by grants from the National Natural Science Foundation of China (81772162 and U1704283) and a grant from the Science and Technology Department of Jilin Province (20230203015SF).

### AUTHOR AFFILIATIONS

[1]Department of Pathogenobiology, Jilin University Mycology Research Center, Key Laboratory of Zoonosis Research, Ministry of Education, College of Basic Medical Sciences, Jilin University, Changchun, China
[2]Department of Dermatology, First Affiliated Hospital of Dalian Medical University, Dalian, China
[3]Beijing ZhongKai TianCheng Bio-technology Co. Ltd., Beijing, China

### AUTHOR ORCIDs

Shumi Shang  http://orcid.org/0009-0000-1876-7937
Li Wang  http://orcid.org/0000-0002-7386-3232

### FUNDING

| Funder | Grant(s) | Author(s) |
| --- | --- | --- |
| MOST | National Natural Science Foundation of China (NSFC) | 81772162 | Li Wang |
| MOST | National Natural Science Foundation of China (NSFC) | U1704283 | Li Wang |

### AUTHOR CONTRIBUTIONS

Shumi Shang, Conceptualization, Formal analysis, Methodology, Writing – original draft | Dan He, Conceptualization, Project administration, Writing – original draft | Cong Liu, Data curation, Methodology | Xinyuan Bao, Formal analysis, Resources | Shuaishuai Han, Software | Li Wang, Writing – review and editing

## ADDITIONAL FILES

The following material is available online.

### Supplemental Material

**Fig. S1 and S2 (Spectrum02699-23-S0001.docx).** Fig. S1: Role of TRAF3 in the interaction between lung epithelial cells and macrophages in A. fumigatus infection. Fig. S2: Effect of TRAF3 to the viability and cell membrane integrity of lung epithelial cells infected with A. fumigatus.

### Open Peer Review

**PEER REVIEW HISTORY (review-history.pdf).** An accounting of the reviewer comments and feedback.

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
