## [Reviewer comments · Microbiology Spectrum]

Microbiology Spectrum

TRAF3 gene regulates macrophage migration and activation by lung epithelial cells infected with *Aspergillus fumigatus*

Shumi Shang, Dan He, Cong Liu, Xinyuan Bao, Shuaishuai Han, and Li Wang

Corresponding Author(s): Li Wang, Jilin University

Review Timeline:

Submission Date:	June 29, 2023
Editorial Decision:	July 27, 2023
Revision Received:	September 11, 2023
Editorial Decision:	October 9, 2023
Revision Received:	October 11, 2023
Accepted:	October 21, 2023

Editor: Lea Atanasova

Reviewer(s): Disclosure of reviewer identity is with reference to reviewer comments included in decision letter(s). The following individuals involved in review of your submission have agreed to reveal their identity: Andrew Wagner (Reviewer #1); Benjamin P. Knox (Reviewer #2)

Transaction Report:

DOI: <https://doi.org/10.1128/spectrum.02699-23>

July 27, 2023

Prof. Li Wang

Jilin University

Pathogenobiology, College of Basic Medical Sciences, Jilin University Mycology Research Center, Jilin University

No.126 Xinmin Street, Changchun City, Jilin Province, China

Changchun, Jilin 130021

China

Re: Spectrum02699-23 (TRAF3 gene regulates macrophage migration and activation by lung epithelial cells infected with *Aspergillus fumigatus*)

Dear Prof. Li Wang:

Thank you for submitting your manuscript to Microbiology Spectrum. Based on the reviewers comments you are kindly invited to submit a revised version of your manuscript. When submitting the revision, please provide (1) point-by-point responses to the issues raised by the reviewers as file type "Response to Reviewers," not in your cover letter, and (2) a PDF file that indicates the changes from the original submission (by highlighting or underlining the changes) as file type "Marked Up Manuscript - For Review Only". Please use this link to submit your revised manuscript - we strongly recommend that you submit your paper within the next 60 days or reach out to me. Detailed instructions on submitting your revised paper are below.

Link Not Available

Sincerely,

Lea Atanasova

Journals Department
Reviewer comments:

Reviewer #1 (Comments for the Author):

General Comments:

-The authors do a nice job showing that TRAF3 overexpression impacts the epithelial response to *A. fumigatus* spores. However, this claim would be strengthened by complimenting overexpression data with knockdown data, or by overexpressing an inhibited TRAF3 gene as a control (possibly a gene lacking the nuclear localization sequence).

-How did the authors differentiate fluorescence between the eGFP signal in transfected A549 cells containing overexpression/control plasmids and the annexin V-FITC fluorescence during microscopy analyses? It is unclear and the two

fluors have nearly identical spectral profiles.

-In figures 2D and 4E, the TRAF3 OV line appears to have more fluorescence than the control in the representative images. As annexin V is a common marker for apoptosis, is it possible that viability of the OV line is altered even without spore addition, and that the down-regulation of inflammatory cytokines in later figures is simply a consequence of toxicity induced by the robust overexpression of this protein? Quantifying Annexin V fluorescence in the absence of spores and complimenting it with a viability or stress dye (like propidium iodide) would be useful in addressing this concern.

-In figure 3, the authors thoroughly showed that TRAF3 overexpression attenuated the production of key pro-inflammatory cytokines by A549 cells in response to fungal spores. Given the positive correlation between TRAF3 activation and type I interferon, it would be interesting to see if IFN-I was inversely increased in response to TRAF3 overexpression and spore stimulation.

-The authors did a nice job looking at signaling pathway activation in figure 5. However, the total levels of these proteins need to be considered as well, as the stoichiometry between activated(phosphorylated)/total protein concentration is a driving factor in these responses. Quantification (like what was done for the phosphorylated states of these proteins) showing if TRAF3 overexpression impacts total p65, I κ B α , ERK and AKT levels should be included to provide justification that overexpression is impacting activation specifically, rather than simply altering the basal levels of these proteins.

-In figure 5C, quantification of nuclear translocation would help strengthen the claim that TRAF3 OV inhibits p65 nuclear translocation in response to spores.

-It would also be interesting to see what happens if fungal cells germinate following internalization. As the TRAF3 OV lines have more internalized spores, it is reasonable to assume that these cells will be more damaged during fungal germination, and consequently may increase pro-inflammatory cytokine production and macrophage recruitment later in the infection process. This information could help explain the hypervirulent phenotype observed in TRAF3 OV transgenic zebrafish as well.

Text Comments:

-In line 63, the authors state "Some studies have shown that *A. fumigatus* infects lung epithelial cells in vitro for 6 hours". Is this specifically referring to the spores prior to germination? Clarifying this statement would be helpful.

-Figures 1A and C on the individual figure 1 PDF and in the image at the end of the compiled manuscript have an X-axis that states that spores are absent in either condition. This should be fixed to show that conditions where TRAF3 expression is reduced have both spores and A549 cells.

-In lines 368-370 in the Immunofluorescence methods section, indicate the concentration or the dilution factor for the antibodies used.

-The statistical analyses performed in each figure are difficult to assess. The authors state that one-way ANOVAs followed by t-tests were used, but many figures show a comparison between only two groups (i.e. figures 1 and 2), where an ANOVA is not appropriate. The authors should more clearly state the type of statistical analysis performed throughout the paper to address this.

Reviewer #2 (Comments for the Author):

Thank you for this submission and clearly telling an interesting story of how TRAF3 regulation in response to *Aspergillus fumigatus* is worthy of greater appreciation and future study.

I have recommended some minor changes. Please see attached comments for details.

Staff Comments:

Preparing Revision Guidelines

Please return the manuscript within 60 days; if you cannot complete the modification within this time period, please contact me. If you do not wish to modify the manuscript and prefer to submit it to another journal, please notify me of your decision immediately so that the manuscript may be formally withdrawn from consideration by Microbiology Spectrum.

The submitted manuscript by Shang *et al.* expands on our understanding of *A. fumigatus* interactions with host lung epithelia by assessing the impact that TRAF3, a key node in immune signaling, has on spore interactions with A549 epithelial cells *in vitro*. The authors nicely showed that TRAF3 expression is decreased in A549 cells in response to fungal spore exposure, which led them to hypothesize that down-regulation of this protein is important in the lung epithelial antifungal response. Accordingly, they showed that ectopic overexpression of TRAF3 increased spore adhesion and internalization by A549 cells and attenuated the production of pro-inflammatory cytokines in response to spore exposure. This corresponded with reduced macrophage recruitment *in vitro*, which they claim is the consequence of greater host immune system evasion via increased A549 internalization, and enhanced susceptibility to *A. fumigatus* infection in transgenic zebrafish lines that overexpressed TRAF3.

Overall, the manuscript is well written, and the experiments are appropriately designed and controlled to answer the questions they sought to ask. However, I have several questions and suggestions that I believe would help strengthen the claims made in this report.

General Comments

-The authors do a nice job showing that TRAF3 overexpression impacts the epithelial response to *A. fumigatus* spores. However, this claim would be strengthened by complimenting overexpression data with knockdown data, or by overexpressing an inhibited TRAF3 gene as a control (possibly a gene lacking the nuclear localization sequence).

-How did the authors differentiate fluorescence between the eGFP signal in transfected A549 cells containing overexpression/control plasmids and the annexin V-FITC fluorescence during microscopy analyses? It is unclear and the two fluors have nearly identical spectral profiles.

-In figures 2D and 4E, the TRAF3 OV line appears to have more fluorescence than the control in the representative images. As annexin V is a common marker for apoptosis, is it possible that viability of the OV line is altered even without spore addition, and that the down-regulation of inflammatory cytokines in later figures is simply a consequence of toxicity induced by the robust overexpression of this protein? Quantifying Annexin V fluorescence in the absence of spores and complimenting it with a viability or stress dye (like propidium iodide) would be useful in addressing this concern.

-In figure 3, the authors thoroughly showed that TRAF3 overexpression attenuated the production of key pro-inflammatory cytokines by A549 cells in response to fungal spores. Given the positive correlation between TRAF3 activation and type I interferon, it would be interesting to see if IFN-I was inversely increased in response to TRAF3 overexpression and spore stimulation.

-The authors did a nice job looking at signaling pathway activation in figure 5. However, the total levels of these proteins need to be considered as well, as the stoichiometry between activated(phosphorylated)/total protein concentrations is a driving factor in these responses.

Quantification (like what was done for the phosphorylated states of these proteins) showing if TRAF3 overexpression impacts total p65, I κ B α , ERK and AKT levels should be included to provide justification that overexpression is impacting activation specifically, rather than simply altering the basal levels of these proteins.

-In figure 5C, quantification of nuclear translocation would help strengthen the claim that TRAF3 OV inhibits p65 nuclear translocation in response to spores.

-It would be interesting to see what happens if fungal cells germinate following internalization. As the TRAF3 OV lines have more internalized spores, it is reasonable to assume that these cells will be more damaged during fungal germination, and consequently may increase pro-inflammatory cytokine production and macrophage recruitment later in the infection process. This information could help explain the hypervirulent phenotype observed in transgenic zebrafish as well.

Text Comment

-In line 63, the authors state “Some studies have shown that *A. fumigatus* infects lung epithelial cells in vitro for 6 hours”. Is this specifically referring to the spores prior to germination? Clarifying this statement would be helpful.

-Figures 1A and C on the individual figure 1 PDF and in the image at the end of the compiled manuscript have an X-axis that states that spores are absent in either condition. This should be fixed to show that conditions where TRAF3 expression is reduced have both spores and A549 cells.

-In lines 368-370 in the Immunofluorescence methods section, indicate the concentration or the dilution factor for the antibodies used.

-The statistical analyses performed in each figure are difficult to assess. The authors state that one-way ANOVAs followed by t-tests were used, but many figures show a comparison between only two groups (i.e. figure 1), where an ANOVA is not appropriate. The authors should more clearly state the type of statistical analysis performed throughout the paper to address this.

2023-07-26

Reviewer Summary:

Here, in “TRAF3 gene regulates macrophage migration and activation by lung epithelial cells infected with *Aspergillus fumigatus*” the authors document a role for TRAF3 in mediating cellular (A549 and THP1) and host (zebrafish) responses to exposure and infection by the filamentous fungus *Aspergillus fumigatus*. Under normal conditions TRAF3 expression and protein abundance goes down in response to *A. fumigatus* spores, likely ensuring appropriate and functional immune response to this opportunistic pathogen. However, disturbing this normal response through upregulation of TRAF3 expression in A549 cells results in increased spore adhesion and internalization, an overall suppression of proinflammatory cytokines, and diminished interaction with alveolar macrophages. In vivo zebrafish data corroborates the in vitro data and further adds a survival component where TRAF3 overexpression results in increased host mortality.

General Comments:

Readers of this work will wonder why TRAF3 is only overexpressed and not knocked down in any of the experimental in vitro and in vivo models here. Therefore, it would be appropriate to acknowledge in the Discussion Section: 1) TRAF3 knockdown/knockout studies in the literature and how those data relate to this study, and 2) why no TRAF3 knockdown/knockout models were used here.

Figure Comments:

Figure 1

- Consider rewording the figure caption: “**FIG1** TRAF3 down-regulation responds to *A. fumigatus* infection in lung epithelial cells”.
- Make non-overlay images in (1D) black/white for contrast and enhance magnification to clearly illustrate co-localization of DAPI and TRAF3 signals.

Figure 2

- Excellent figure caption title
- Same image comments as for Fig.1 – make non-overlay images black/white for contrast and only use color for overlays. Consider also changing the colors (for example, DAPI doesn't have to be dark blue) on overlay for enhanced contrast.

Figure 3

- Excellent figure caption title
- Using “-“ to represent “normal” TRAF3 expression is misleading as readers are more likely to interpret it as knockdown or knockout. Please use different nomenclature for TRAF3 than +/- to avoid confusion. Especially because “+/-“ is used accurately to represent “presence/absence” of spores.

Figure 4

- Update caption title to describe the figure's data and summary takeaway.
- Similar comment as in Fig 3 – use different nomenclature other than (-) to indicate normal/control TRAF3 levels.
- Fig 4D – please ensure figure graph legend is clear: “A549-TRAF3” suggests at a quick glance a loss of TRAF3 when in fact it is being overexpressed. Consider, for example, “A549+TRAF3” which would still work in the other condition with spores “A549+TRAF3+Spore”
- Same image comments as before.

Figure 5

- Single channel images should be black and white for contrast. Overlays can be color.
- p-ERK and p-AKT Western (5D) and quantitation (5E) data do not seem to visually agree with each other. Perhaps the blot was overexposed? Is there a better representative image? Was p-ERK and p-AKT normalized to ERK and AKT, respectively, and not GAPDH? These discrepancies between 5D and 5E undermine confident takeaways of the conclusions made from this figure.

Figure 6

- Update caption title to describe the figure's data and summary takeaway.
- Show TRAF3 relative mRNA in zebrafish uninfected with *A. fumigatus* spores to demonstrate overexpression *in vivo*. This could be supplemental data.

Figure 7

- Excellent schematic that summarizes the paper's data

Supplementary Figure 1

- Caption title should summarize data and takeaway
- This is a great insight.

Thank you for the reviewers' comments concerning our manuscript entitled "TRAF3 gene regulates macrophage migration and activation by lung epithelial cells infected with *Aspergillus fumigatus*" (ID: Spectrum02699-23). We appreciate the time and effort each of the reviewers have dedicated to providing feedback on our manuscript and are grateful for the insightful comments and valuable advice to our paper. We have studied comments carefully and we did our best to reply to the comments and we hope that it will meet your approval. Revised portions are marked in purple in the paper, for a point-by-point response to the reviewers' comments and concerns.

Responds to the reviewer's comments:

Reviewer #1

General Comments

-The authors do a nice job showing that TRAF3 overexpression impacts the epithelial response to *A. fumigatus* spores. However, this claim would be strengthened by complementing overexpression data with knockdown data, or by overexpressing an inhibited TRAF3 gene as a control (possibly a gene lacking the nuclear localization sequence).

Response: We agree with the comment that raised an important topic. TRAF3 knockout mice die in the early stages of life, indicating that this gene has an important biological function in postnatal development and in maintaining normal immune system function. Moreover, there are no agonists or inhibitors available to investigate the TRAF3 gene, which makes the study of the biological function of TRAF3 more difficult. First of all, in this research we found that TRAF3 gene expression was down-regulated after infection of lung epithelial cells with *A. fumigatus*. Considering also the difficulty of animal model construction and the consistency of in vivo and in vitro experiments. In this study, we chose to construct an in vivo and in vitro model of TRAF3 gene overexpression to in turn demonstrate the role played by TRAF3 gene in the early infection process of *A. fumigatus*, and added this idea to the discussion section of the article(see line 213-220).

-How did the authors differentiate fluorescence between the eGFP signal in transfected A549 cells containing overexpression/control plasmids and the annexin V-FITC fluorescence during microscopy analyses? It is unclear and the two fluorophores have nearly identical spectral profiles.

Response: Thank you for pointing this out. When we designed the adhesion experiment, the first choice was the spontaneous eGFP fluorescence after cell transfection with the plasmid. But after cell fixation, the eGFP fluorescence was weak, and we needed to increase the exposure time to see the fluorescence clearly, which made the cell fluorescence counting inaccurate. We wanted to find a universal fluorescent dye for labeling cell membranes. And then we associated that Annexin V can label the cell membrane of dead cells, while cells die after fixation. Therefore we added Annexin V staining of cell membranes, which exacerbated the fluorescence of the membranes and allowed us to see clearly the membrane boundaries in a shorter exposure time. However, your comment suggests us that Annexin V and eGFP have the same green fluorescence, which is indeed not easy to distinguish, and it is also easy to cause confusion in the article. So, we chose Annexin V-PE dye to label the cell membrane and repeated the experiment (see Figure 2D, 4E). And we modified the materials and methods involving adhesion experiments(see line 322).

-In figures 2D and 4E, the TRAF3 OV line appears to have more fluorescence than the control in the representative images. As annexin V is a common marker for apoptosis, is it possible that viability of the OV line is altered even without spore addition, and that the down-regulation of inflammatory cytokines in later figures is simply a consequence of toxicity induced by the robust overexpression of this protein? Quantifying Annexin V fluorescence in the absence of spores and complimenting it with a viability or stress dye (like propidium iodide) would be useful in addressing this concern.

Response: We agree with the reviewer's comment. We added Supplementary Figure 2 in the manuscript to detect lung epithelial cell viability and cell membrane integrity by MTT assay and LDH release assay. This result showed that overexpression of TRAF3 gene in lung epithelial cells did not affect cell viability changes and cell membrane integrity in the absence of *A. fumigatus* spores(see line 145-152). Therefore, it can be speculated that the down-regulation of pro-inflammatory cytokine expression is not related to the consequences of toxicity induced by the robust overexpression of the TRAF3 protein (see Supplementary Figure 2A, B).

-In figure 3, the authors thoroughly showed that TRAF3 overexpression attenuated the production of key pro-inflammatory cytokines by A549 cells in response to fungal spores. Given the positive correlation between TRAF3 activation and type I interferon, it would be interesting to see if IFN-I was inversely increased in response to TRAF3 overexpression and spore stimulation.

Response: Thank you for pointing this out. Type I IFN is mainly composed of two types, IFN- α and IFN- β . During the experimental process of this study, we also detected the expression of IFN- α and IFN- β , but their expression was very low in A549 cells, and was difficult to be detected. By reviewing the literature, we found that the expression of type I IFN is induced by viral attack. And, IFN-I is mainly produced by fibroblasts and leukocytes. These points can support our qPCR results, so the role played by IFN-I in TRAF3 overexpression and spore stimulation was not considered in this study.

References:

- Pestka S, Krause CD, Walter MR. Interferons, interferon-like cytokines, and their

receptors. *Immunol Rev.* 2004 Dec;202:8-32. doi: 10.1111/j.0105-2896.2004.00204.x. PMID: 15546383.

- Zhao M, Li J, Ji H, Chen D, Hu H. A versatile endosome acidity-induced sheddable gene delivery system: increased tumor targeting and enhanced transfection efficiency. *Int J Nanomedicine.* 2019 Aug 14;14:6519-6538. doi: 10.2147/IJN.S215250. PMID: 31616142; PMCID: PMC6698616.

-The authors did a nice job looking at signaling pathway activation in figure 5. However, the total levels of these proteins need to be considered as well, as the stoichiometry between activated(phosphorylated)/total protein concentrations is a driving factor in these responses. Quantification (like what was done for the phosphorylated states of these proteins) showing if TRAF3 overexpression impacts total p65, IκBα, ERK and AKT levels should be included to provide justification that overexpression is impacting activation specifically, rather than simply altering the basal levels of these proteins.

Response:We agree with the reviewers' comments. Therefore, we modified the grayscale value analysis in Figure 5B, G. The total levels of p65, IκBα, ERK, and AKT were quantified in Figure 5A, F. The ratio of phosphorylated protein value to total protein value was employed to express the changes in protein activation (see Figure 5B, G).

-In figure 5C, quantification of nuclear translocation would help strengthen the claim that TRAF3 OV inhibits p65 nuclear translocation in response to spores.

Response: We agree with the reviewers' comments. Therefore, we supplemented Figure 5C, D. Proteins were extracted from the nucleus and cytoplasm, separately, and their p65 protein expression was detected by Western blot to quantify the nuclear displacement of p65 (see Figure 5C, D) to clarify the concerns of the reviewers.

-It would be interesting to see what happens if fungal cells germinate following internalization. As the TRAF3 OV lines have more internalized spores, it is reasonable to assume that these cells will be more damaged during fungal germination, and consequently may increase pro-inflammatory cytokine production and macrophage recruitment later in the infection process. This information could help explain the hypervirulent phenotype observed in transgenic zebrafish as well.

Response: Thank you for pointing this out. We have added Supplementary Figure 2 to the manuscript to verified the effect of TRAF3 overexpression subsequent to the viability and cell damage of *A. fumigatus*-infected lung epithelial cells. The results showed that TRAF3 did not significantly impact the alteration of A549 cell viability induced by *A. fumigatus*. Therefore, it can be assumed that TRAF3 inhibition of *A. fumigatus*-induced pro-inflammatory cytokine production is not related to cell injury(see line 145-152).

The results of the previous study of our team showed that *Aspergillus fumigatus* started to germinate at 6h when cultured in vitro. When *A. fumigatus* was co-cultured with human umbilical vein endothelial cells (HUVEC), HUVEC was able to increase the germination time of *A. fumigatus* spores to 8 h. The first events that occur when *A. fumigatus* spores infect the host are adhesion and internalization. Combining the literature and the previous research results of our team, we believe that the main events occurring within 6h of *A. fumigatus* infecting cells in vitro are adhesion and

internalization, which is the main research content of our manuscript.

As follows:

-Zhang W, He D, Wei Y, Shang S, Li D, Wang L. Suppression of *Aspergillus fumigatus* Germination by Neutrophils Is Enhanced by Endothelial-Derived CSF3 Production. *Front Microbiol.* 2022 Apr 29;13:837776. doi: 10.3389/fmicb.2022.837776. PMID: 35572651; PMCID: PMC9100814.

Text Comment

-In line 63, the authors state “Some studies have shown that *A. fumigatus* infects lung epithelial cells in vitro for 6 hours”. Is this specifically referring to the spores prior to germination? Clarifying this statement would be helpful.

Response: Thank you for your advice. We have made additions here. It is clarified that when *A. fumigatus* infects lung epithelial cells in vitro, it first adheres to the cell surface and then internalizes into the cell, with the event that *A. fumigatus* spores begin to germinate 6 hours after infection(see line 62-64).

-Figures 1A and C on the individual figure 1 PDF and in the image at the end of the compiled manuscript have an X-axis that states that spores are absent in either condition. This should be fixed to show that conditions where TRAF3 expression is reduced have both spores and A549 cells.

Response: Thank you for pointing this out. This was an oversight in our uploading and have re-uploaded the correct figure 1.

-In lines 368-370 in the Immunofluorescence methods section, indicate the concentration or the dilution factor for the antibodies used.

Response: Thank you for pointing this out. We have made detailed additions to the dilutions of the antibodies used for the Wbtern blot, immunofluorescence experiments in the Materials and Methods section of the manuscript(see line 353-357 and line 393-395).

-The statistical analyses performed in each figure are difficult to assess. The authors state that one-way ANOVAs followed by t-tests were used, but many figures show a comparison between only two groups (i.e. figure 1), where an ANOVA is not appropriate. The authors should more clearly state the type of statistical analysis performed throughout the paper to address this.

Response: Thank you for pointing this out. We re-analyzed the comparisons between the two groups using unpaired two-tailed *t*-tests (see Figures 1A, C, and E, Figure 2A, C, E, and F). And, we supplemented the statistical analysis methods in the manuscript(see line 423-425).

Reviewer #2

General Comments:

Readers of this work will wonder why TRAF3 is only overexpressed and not knocked down in any of the experimental in vitro and in vivo models here. Therefore, it would be appropriate to acknowledge in the Discussion Section: 1) TRAF3 knockdown/knockout studies in the literature and how those data relate to this study, and 2) why no TRAF3 knockdown/knockout models were used here.

Response: We agree that the review raises an important issue. Referring to the reviewer's comments and taking into account the published literature, we have added in the Discussion section of the manuscript additional reasons why the TRAF3 knockout model was not used in this study. TRAF3 knockout mice die in the early stages of life, indicating that this gene has an important biological function in postnatal development and in maintaining normal immune system function. Moreover, there are no agonists or inhibitors available to investigate the TRAF3 gene, which makes the study of the biological function of TRAF3 more difficult. First of all, in this research we found that TRAF3 gene expression was down-regulated after infection of lung epithelial cells with *A. fumigatus*. Considering also the difficulty of animal model construction and the consistency of in vivo and in vitro experiments. In this study, we chose to construct an in vivo and in vitro model of TRAF3 gene overexpression to in turn demonstrate the role played by TRAF3 gene in the early infection process of *A. fumigatus*(see line 213-220).

Figure 1

- Consider rewording the figure caption: "FIG1 TRAF3 down-regulation responds to *A. fumigatus* infection in lung epithelial cells".
- Make non-overlay images in (1D) black/white for contrast and enhance magnification to clearly illustrate co-localization of DAPI and TRAF3 signals.

Response:

•Thank you for pointing this out. We have changed the title(see line 103), " FIG1 TRAF3 down-regulation responds to *A. fumigatus* infection in lung epithelial cells," to read, " FIG1 Lung epithelial cell TRAF3 expression is down-regulated under *A. fumigatus* infection." If it looks better this way?

•Thank you for your advice. This comment can suggest that non-overlaid black/white images help increase the contrast between different fluorescences. So, we followed your suggestion and tried to change 1D to a black/white image (see Figure II below), and there was little difference in the changes shown with the original image (see Figure I below) for TRAF3 expression. Therefore, we did not change the color scheme of the original image and only supplemented the local zoomed image to show the co-localization of DAPI and TRAF3 signals. Meanwhile, we supplemented Figure 1E with the average fluorescence intensity calculated using image J software to show the differences in TRAF3 expression (see Figure 1E).

FIG. TRAF3 expression and localization in lung epithelial cells.

(I) Figure 1D in the original text; (II) TRAF3 fluorescence signals were converted to black/white color matching and merged.

Figure 2

- Excellent figure caption title
- Same image comments as for Fig.1 – make non-overlay images black/white for contrast and only use color for overlays. Consider also changing the colors (for example, DAPI doesn't have to be dark blue) on overlay for enhanced contrast.

Response:

- Thank you for your affirmation.
- Thank you for your advice. We set the Figure 2D image to black/white, but we do not think the difference is significant (after the image conversion, the contrast of the effect is similar to the comparison image placed above, so we will not show it here). Considering that we applied image J software to count the number of cells and spores (see Figure 2E), thus no changes were made to the original image.

Figure 3

- Excellent figure caption title
- Using “-“ to represent “normal” TRAF3 expression is misleading as readers are more likely to interpret it as knockdown or knockout. Please use different nomenclature for TRAF3 than +/- to avoid confusion. Especially because “+/-“ is used accurately to

represent "presence/absence" of spores.

Response:

- Thank you for your affirmation.
- We agree with the reviewers' comments. We chose "Vector" to represent A549 cells transfected with the control vector plasmid and "TRAF3" to represent A549 cells transfected with the TRAF3 overexpression plasmid (see Figure 3).

Figure 4

- Update caption title to describe the figure's data and summary takeaway.
- Similar comment as in Fig 3 – use different nomenclature other than (-) to indicate normal/control TRAF3 levels.
- Fig 4D – please ensure figure graph legend is clear: "A549-TRAF3" suggests at a quick glance a loss of TRAF3 when in fact it is being overexpressed. Consider, for example, "A549+TRAF3" which would still work in the other condition with spores "A549+TRAF3+Spore"
- Same image comments as before.

Response:

- Thank you for your comments. We changed the title of Figure 4 from "Role of TRAF3 in the interaction between lung epithelial cells and macrophages in *A. fumigatus* infection." to "TRAF3 inhibits macrophage migration and macrophage cytokine production by lung epithelial cells infected with *A. fumigatus*.". If it looks better this way?
- Thank you for pointing this out. We chose "Vector" to represent A549 cells transfected with the control vector plasmid and "TRAF3" to represent A549 cells transfected with the TRAF3 overexpression plasmid (see Figure 4).
- We agree with the reviewers' comments. We chose "Vector" to represent A549 cells transfected with the control vector plasmid and "TRAF3" to represent A549 cells transfected with the TRAF3 overexpression plasmid. Meanwhile, we changed the legend "A549+Spore", "A549-TRAF3+Spore" to "Vector+Spore", "TRAF3+Spore" for the presence of spores(see Figure 4D).
- Thanks to the reviewers for their comments. We tried changing the fluorescence image of 4E to black/white to enhance the contrast, but we still think the original image better illustrates the phenomenon(after the image conversion, the contrast of the effect is similar to the comparison image placed above, so we will not show it here). Meanwhile, considering that we applied image J software to count the number of cells and spores, we did not change the original figure(see Figure 4F).

Figure 5

- Single channel images should be black and white for contrast. Overlays can be color.
- p-ERK and p-AKT Western (5D) and quantitation (5E) data do not seem to visually agree with each other. Perhaps the blot was overexposed? Is there a better representative image? Was p-ERK and p-AKT normalized to ERK and AKT, respectively, and not GAPDH? These discrepancies between 5D and 5E undermine confident takeaways of the conclusions made from this figure.

Response:

- Thank you to the reviewers comment. We tried to change the fluorescence picture of

5E to black/white to enhance the contrast, but we still think the original picture better illustrates the phenomenon. Furthermore, we added Western blot experiments of protein expression of p65 in the nucleus and cytoplasm (see Figure 5C and D), which supplemented to illustrate that TRAF3 was overexpressed in A549 cells, the accumulation of p65 in the nucleus of A549 cells induced by stimulation of *A. fumigatus* was significantly reduced.

- We agree with the reviewer's comments. We chose to replace Figure 5D (see Figure 5F) with another set of Western blot images from the duplicate experiments. We also quantified the total levels of p65, IκBα, ERK, and AKT in Figure 5, and the ratio of phosphorylation/total protein concentration was also chosen to indicate the degree of protein activation in the grayscale value analysis (see Figure 5B, F, G).

Figure 6

- Update caption title to describe the figure's data and summary takeaway.
- Show TRAF3 relative mRNA in zebrafish uninfected with *A. fumigatus* spores to demonstrate overexpression in vivo. This could be supplemental data.

Response:

- Thank you for your advice. We have changed the title of Figure 6 from "Role of TRAF3 in resistance to *A. fumigatus* infection in zebrafish." to "TRAF3 overexpression reduces survival and promotes fungal load in vivo in *A. fumigatus*-infected zebrafish.". If it looks better this way?

- We agree with the reviewer's comments. The mRNA expression of TRAF3 in zebrafish uninfected with *A. fumigatus* spores is supplemented in Figure 6B to demonstrate the successful construction of transgenic zebrafish overexpressing the TRAF3 gene (see Figure 6B).

Figure 7

- Excellent schematic that summarizes the paper's data

Response:

- Thank you for your affirmation.

Supplementary Figure 1

- Caption title should summarize data and takeaway
- This is a great insight

Response:

- Thank you for your advice. We have changed the title of Supplementary Figure 1 from "Role of TRAF3 in the interaction between lung epithelial cells and macrophages in *A. fumigatus* infection." to "Medium from *A. fumigatus*-infected lung epithelial cells does not induce macrophage migration". Will it look better this way?

- Thank you for your affirmation.

October 9, 2023

Prof. Li Wang
Jilin University
Pathogenobiology, College of Basic Medical Sciences, Jilin University Mycology Research Center, Jilin University
No.126 Xinmin Street, Changchun City, Jilin Province, China
Changchun, Jilin 130021
China

Re: Spectrum02699-23R1 (TRAF3 gene regulates macrophage migration and activation by lung epithelial cells infected with *Aspergillus fumigatus*)

Dear Prof. Li Wang:

Thank you for submitting your manuscript to Microbiology Spectrum. As you will see your paper is very close to acceptance. Please reply to the question that one of the reviewer is posing. As these revisions are quite minor, I expect that you should be able to turn in the revised paper in less than 30 days, if not sooner. You will find the reviewers' comments below.

When submitting the revised version of your paper, please provide (1) point-by-point responses to the issues raised by the reviewers as file type "Response to Reviewers," not in your cover letter, and (2) a PDF file that indicates the changes from the original submission (by highlighting or underlining the changes) as file type "Marked Up Manuscript - For Review Only". Please use this link to submit your revised manuscript. Detailed instructions on submitting your revised paper are below.

Link Not Available

Sincerely,
Lea Atanasova
Editor, Microbiology Spectrum

Reviewer comments:

Reviewer #1 (Comments for the Author):

The authors do a very nice job addressing all previous and comments and concerns. Overall, I have no more major suggestions and think the authors have done nicely to put a strong manuscript together. However, I do have one final question that needs clarification. In supplementary figure 2, the results in 2A show that there is a significant difference in viability in lines that overexpress TRAF3. However, it appears that in this condition the vector only control did not receive any spores in either experimental group, but the TRAF3 overexpression strain did receive spores in both. Could there be a labelling error here? The same is true for Fig S2B. If the labels are correct, then this suggests that there is a difference in viability when TRAF3 is overexpressed.

Reviewer #2 (Comments for the Author):

Thanks to all the contributing authors on this work for revising an improved manuscript. It has been a joy for me to be part of this scientific dialogue and I wish you all much success with this paper and beyond.

Preparing Revision Guidelines

To submit your modified manuscript, log onto the eJP submission site at <https://spectrum.msubmit.net/cgi-bin/main.plex>. Go to

Author Tasks and click the appropriate manuscript title to begin the revision process. The information that you entered when you first submitted the paper will be displayed. Please update the information as necessary. Here are a few examples of required updates that authors must address:

Please return the manuscript within 60 days; if you cannot complete the modification within this time period, please contact me. If you do not wish to modify the manuscript and prefer to submit it to another journal, please notify me of your decision immediately so that the manuscript may be formally withdrawn from consideration by Microbiology Spectrum.

The authors do a very nice job addressing all previous and comments and concerns. Overall, I have no more major suggestions and think the authors have done nicely to put a strong manuscript together. However, I do have one final question that needs clarification:

Supplementary Figure 2. Effect of TRAF3 to the viability and cell membrane integrity of lung epithelial cells infected with *A. fumigatus* (A) Viability of A549 cells was detected by MTT conversion. (B) Detection of A549 cell membrane integrity by LDH release.

In supplementary figure 2, the results in 2A show that there is a significant difference in viability in lines that overexpress *TRAF3*. However, it appears that in this condition the vector only control did not receive any spores in either experimental group, but the *TRAF3* overexpression strain did receive spores in both. Could there be a labelling error here? The same is true for Fig S2B. If the labels are correct, then this suggests that there is a difference in viability when *TRAF3* is overexpressed.

2023-09-18

Reviewer Comments to Revision:

Thank you for entertaining my previous suggestions.

I really enjoy the zoom out panels used in Figure 1! Those are beautiful images and the enhanced detail afforded by the magnification is appreciated and supports the claims of the figure.

However, I must apologize for a lack of clarity in one of my earlier suggestions regarding image presentation for enhanced contrast. Your response to that comment, while appreciated, suggests an error on my part in not being clear with the instructions. The following suggestion is only for consideration in future data visualization and not tied to this resubmission.

- Each fluorescence channel, when shown individually, should be black and white.
 - For example, in Fig. 1: TRAF3 (green) and DAPI (blue) single channel images could BOTH be shown in black and white for enhanced contrast. In single-channel images it is difficult for the human eye to appreciate full contrast between black and dark green or blue.
- Overlay images (green + blue), on the other hand, are shown in their respective colors. When >1 color is present, contrast and overlay is easier to assess over a black background.

Thank you for considering my suggestions on image formatting and visualization. As a viewer/reader I always appreciate all single channel images in grayscale and overlays being the only case where color is used.

We sincerely appreciate to the editor and all reviewers for their valuable feedback on our manuscript entitled "TRAF3 gene regulates macrophage migration and activation by lung epithelial cells infected with *Aspergillus fumigatus*" (ID: Spectrum02699-23), which we used to improve the quality of our manuscript. As suggested by the reviewers' comments, we revised Supplementary Fig. S2 and unified the representation of the statistical graphs in Fig. 3, Fig. 4, and Fig. S1. Since we revised the figures this time, we only uploaded the Manuscript: A .DOC version of the revised manuscript and the revised figures, and did not re-upload the "Marked-Up Manuscript" file (without figures). Below is our response to this question from the reviewers.

Reviewer #1 (Comments for the Author):

The authors do a very nice job addressing all previous and comments and concerns. Overall, I have no more major suggestions and think the authors have done nicely to put a strong manuscript together. However, I do have one final question that needs clarification. In supplementary figure 2, the results in 2A show that there is a significant difference in viability in lines that overexpress TRAF3. However, it appears that in this condition the vector only control did not receive any spores in either experimental group, but the TRAF3 overexpression strain did receive spores in both. Could there be a labelling error here? The same is true for Fig S2B. If the labels are correct, then this suggests that there is a difference in viability when TRAF3 is overexpressed.

Response: We are really very sorry for our careless mistakes. And we sincerely thank the reviewer for careful reading and pointing out this critical error for us. In Supplementary Fig. 2, we have indeed set up four groups "Vector, Vector+Spore, TRAF3, TRAF3+Spore", and the description of the results in the manuscript is based on the results of these four groups (see lines 145-152). In Revision 1, we changed the expression of A549 cells transfected with the empty vector plasmid, TRAF3 overexpression plasmid in the figure. However, due to an oversight on our part, we set the wrong color markers for the subgroups in the statistical graphs. We corrected Supplementary Fig. 2 and double-checked the manuscript to revise the representation of the statistical graphs in Fig. 3, Fig. 4, and Fig. S1 in a uniform manner. After revision, Supplementary Fig. 2 was able to illustrate that TRAF3 gene overexpression in A549 cells did not affect cell viability and cell membrane integrity. Thanks again for the reminder, which saved us a large loss.

Re: Spectrum02699-23R2 (TRAF3 gene regulates macrophage migration and activation by lung epithelial cells infected with *Aspergillus fumigatus*)

Dear Prof. Li Wang:

I am happy to inform you that your manuscript has now been accepted, and I am forwarding it to the ASM production staff for publication. Your paper will first be checked to make sure all elements meet the technical requirements. ASM staff will contact you if anything needs to be revised before copyediting and production can begin. Otherwise, you will be notified when your proofs are ready to be viewed.

Sincerely,
Lea Atanasova
Editor
Microbiology Spectrum